# RSKDD-Net: Random Sample-based Keypoint Detector and Descriptor

**Fan Lu**
Tongji University
*lufan@tongji.edu.cn*

**Guang Chen**[*]
Tongji University
*guangchen@tongji.edu.cn*

**Yinlong Liu**
Technische Universität München
*Yinlong.Liu@tum.de*

**Zhongnan Qu**
ETH Zurich
*quz@ethz.ch*

**Alois Knoll**
Technische Universität München
*knoll@in.tum.de*

## Abstract

Keypoint detector and descriptor are two main components of point cloud registration. Previous learning-based keypoint detectors rely on saliency estimation for each point or farthest point sample (FPS) for candidate points selection, which are inefficient and not applicable in large scale scenes. This paper proposes Random Sample-based Keypoint Detector and Descriptor Network (RSKDD-Net) for large scale point cloud registration. The key idea is using random sampling to efficiently select candidate points and using a learning-based method to jointly generate keypoints and descriptors. To tackle the information loss of random sampling, we exploit a novel random dilation cluster strategy to enlarge the receptive field of each sampled point and an attention mechanism to aggregate the positions and features of neighbor points. Furthermore, we propose a matching loss to train the descriptor in a weakly supervised manner. Extensive experiments on two large scale outdoor LiDAR datasets show that the proposed RSKDD-Net achieves state-of-the-art performance with more than 15 times faster than existing methods. Our code is available at `https://github.com/ispc-lab/RSKDD-Net`.

## 1 Introduction

Point cloud registration is an important problem in 3D computer vision, which aims to estimate the optimal rigid transformation between two point clouds. 3D keypoint detection and description are two fundamental components of point cloud registration. Inspired by numerous handcrafted 2D keypoint detectors and descriptors [1–3], researchers proposed several handcrafted 3D keypoint detectors [4–6] and descriptors [7–11] for point cloud. However, additional RGB channels in images contain richer information than Euclidean coordinates of point cloud without RGB information, which makes handcrafted 3D keypoint detectors and descriptors less reliable than that in 2D images.

With the rapid development of deep learning, many works have explored learning-based methods for 3D descriptors in point cloud [12–15]. However, only a few works explore deep learning-based methods in 3D keypoint detection due to the lack of ground truth dataset for keypoint detector [16]. 3DFeatNet[17] and USIP[16] are two pioneering works of learning-based keypoint detectors, however, they have less efficiency consideration. 3DFeatNet predicts saliency for each input point and select keypoints based on the predicted saliency. The per-point saliency estimation requires a considerable time thus is not applicable in practice. USIP relies on FPS to generate keypoint candidates. However, FPS has a time complexity of $\mathcal{O}(N^2)$, therefore is inefficient and time-consuming. Hence, both two

---

[*]Corresponding author

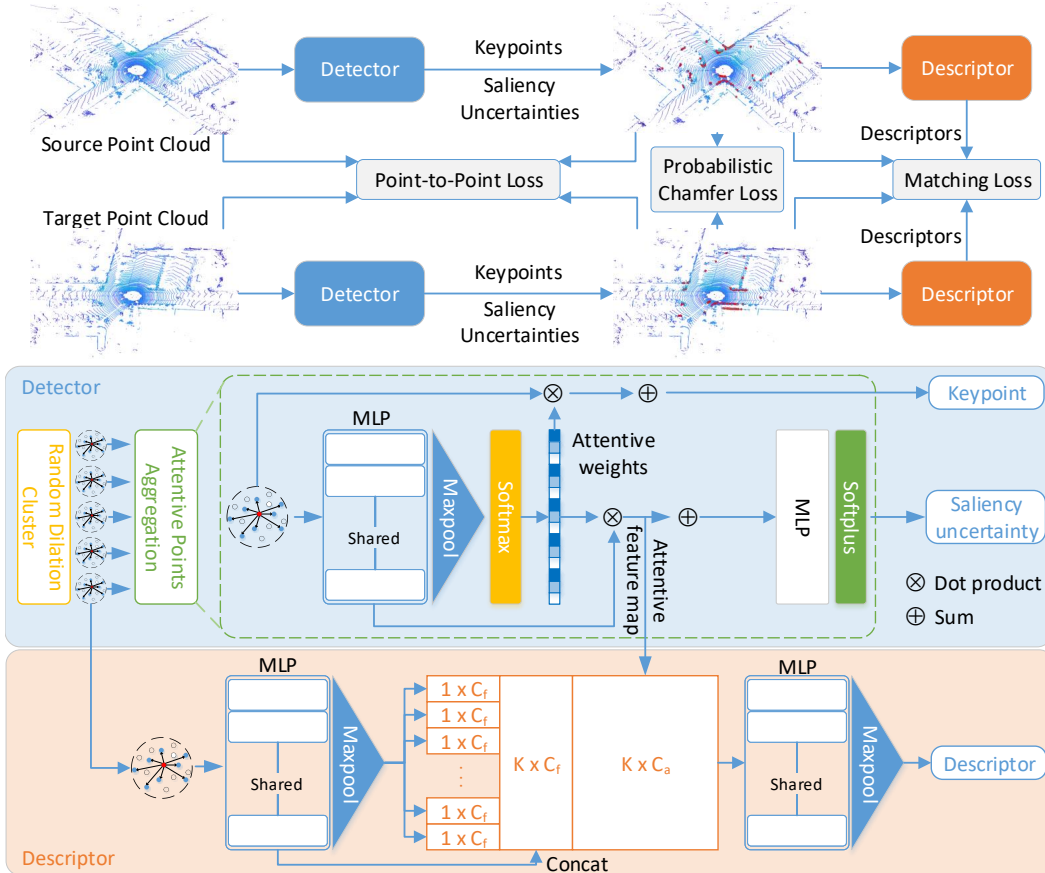

Figure 1: Network architecture of our proposed RSKDD-Net. The top row is the overall structure of the whole network, the middle row is detector network and the bottom row is descriptor network.

methods above can not efficiently process large scale point clouds, which restricts their applications in scenes that require real-time performance, such as autonomous driving.

Based on the above observation, we propose our network named as Random Sample-based Keypoint Detector and Descriptor Network (RSKDD-Net), which jointly generates keypoints and the corresponding descriptors for large scale point cloud efficiently. In this paper, we introduce the random sample concept in RSKDD-Net to improve the efficiency of our network, which is ill-considered in 3DFeatNet and USIP. Random sampling is highly efficient and has been utilized in point cloud semantic segmentation to improve the efficiency [18], however, can lead to a considerable information loss. Inspired by the success of dilation strategy in deep learning of 2D image [19], we propose a novel random dilation strategy to cluster neighbor points, which significantly enlarges the receptive field. We exploit attention mechanism to aggregate the positions and features of neighbor points to generate keypoints and also an attentive feature map to estimate saliency uncertainty of each keypoint. The generative framework avoids inefficient per-point saliency estimation in 3DFeatNet [17]. To jointly learn keypoint detector and descriptor, the clusters and attentive feature maps are further fed into the descriptor network to generate descriptors. To train the descriptor in a weakly supervised manner, we introduce the matching loss, which utilizes a soft assignment strategy to estimate correspondences of keypoints. The network architecture can be seen in Fig. 1.

Extensive experiments are performed to evaluate our RSKDD-Net. The results show that our approach achieves state-of-the-art performance with much lower computation time.

**Contribution** To summarize, our main contributions are as follows:

- We propose a deep learning-based method to jointly detect keypoints and generate descriptors for large scale point cloud registration. The proposed method achieves state-of-the-art performance with a more than $15\times$ higher speed.

- We propose a novel random dilation strategy to enlarge the receptive field, which significantly improves the performance of keypoint detector and descriptor. Besides, an attention mechanism is introduced to aggregate the positions and features of neighbor points.
- We propose an effective matching loss based on soft assignment strategy so that the descriptor can be trained in a weakly supervised manner.

## 2 Related work

Existing approaches of keypoint detector and descriptor for point cloud can be categorized into handcrafted and learning-based approaches.

**Handcrafted approaches** The current handcrafted 3D keypoint detectors and descriptors are mainly inspired by numerous handcrafted methods in 2D images. SIFT-3D [5] and Harris-3D [6] are 3D extensions of widely used 2D detectors SIFT [2] and Harris [3]. Intrinsic Shape Signatures (ISS) [4] selects points where the neighbor points in a ball region have large variations along each principal axis. For the description of keypoints, researchers have also developed several 3D descriptors based on the geometric features of points, like Point Feature Histograms (PFH) [7], Fast Point Feature Histograms (FPFH) [8] and Signature of Histograms of Orientations (SHOT) [9]. A comprehensive introduction of handcrafted 3D detectors and descriptors can be found in [20, 21].

**Learning-based approaches** Recent years, deep learning-based methods have been widely used for point cloud analysis [22–29]. The most relevant approaches to our work are 3DFeatNet [17] and USIP [16]. Unlike previous learning-based descriptors [12] which rely on ground truth matched pairs to train the network, 3DFeatNet proposed a weakly supervised 3D descriptor. The network samples positive and negative pairs according to the distance of point cloud and utilizes a triplet network to train the descriptor. For keypoints detection, they simply predict attentive weight for each point in point cloud and select salient points without more precise optimization. Unlike 3DFeatNet, the focus point of USIP is keypoint detector and how to train the detector fully unsupervised. They sample keypoint candidates using FPS and use SOM [25] to organize the point cloud. An offset to the original candidate points and saliency uncertainty are predicted to select keypoints. USIP proposes probabilistic chamfer loss and point-to-point loss to train the network fully unsupervised. However, saliency estimation for each point of 3DFeatNet and FPS in USIP are both inefficient.

## 3 Approach

The network architecture of our proposed RSKDD-Net is illustrated in Fig. 1. The input point cloud $\mathcal{P} \in \mathbb{R}^{N \times (3+C)}$ (3D Euclidean coordinates and $C$ additional channels) is firstly fed into the detector network. Random dilation cluster is utilized to cluster neighbor points and then an attentive points aggregation module is followed to generate keypoints $\mathbf{X} \in \mathbb{R}^{M \times 3}$, saliency uncertainties $\mathbf{\Sigma} \in \mathbb{R}^M$ and attentive feature maps. The clusters and attentive feature maps are further fed into the descriptor network to generate corresponding descriptors. We train the detector network with probabilistic chamfer loss and point-to-point loss and the descriptor network with the proposed matching loss.

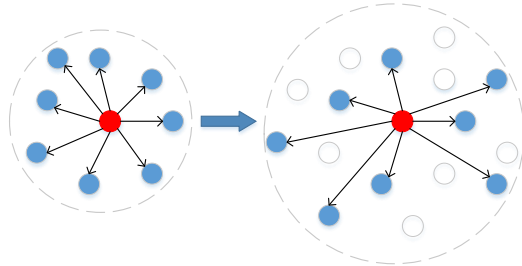

Figure 2: The illustration of our random dilation cluster strategy. The red point is the center point and the blue points are the selected neighbor points. The left part is the standard $k$NN-based cluster and the right part is our random dilation cluster. It is obvious that our method provides a significant enlargement of the receptive field.

### 3.1 Detector

**Random dilation cluster** For each input point cloud $\mathcal{P} \in \mathbb{R}^{N \times (3+C)}$, we firstly random sample $M$ candidate points. Generally, $k$ nearest neighbor ($k$NN) search is performed to group $M$ clusters centered on the candidate points. Although efficient, random sampling can lead to an information loss and expanding the receptive

field is an effective strategy to weaken the negative effect. However, simply enlarging the number of neighbor points $K$ will bring an increase of time and space complexity. Dilation is an alternative strategy to enlarge the receptive field in numerous fields [30, 19]. [31] introduces dilated point convolutions (DPC) in 3D point cloud. Unlike DPC, here we adopt a random dilation strategy to generate clusters and the visual interpretation of this method is shown in Fig. 2. Suppose that $K$ neighbor points are selected for a single cluster, and the dilation ratio is denoted as $\alpha_d$. We firstly search $\alpha_d \times K$ neighbor points of the center point and then random sample $K$ points from them. The proposed strategy is simple however effective, which expands the receptive field from $K$ to $\alpha_d \times K$ with almost no additional computational cost.

**Attentive points aggregation** Attention mechanism has achieved great performance in point cloud learning [32–36]. In this paper, we propose a simple attention mechanism to aggregate neighbor points to directly generate keypoint rather than predicting an offset like USIP [16]. From random dilation cluster, we obtain $M$ clusters and each cluster consists of $K$ points. Considering one cluster, where the center point and $K$ neighbor points are denoted as $p^i$ and $\{p^i_1, \cdots, p^i_k, \cdots, p^i_K\} \in \mathbb{R}^{K \times (3+C)}$, respectively. Euclidean coordinates of neighbor points are subtracted by the center point to get the relative position of each neighbor point. Besides, the relative distances of neighbor points to the center point are calculated as an additional channel of the cluster. Consequently, the feature of a single cluster is denoted as $F^i = \{f^i_1, \cdots, f^i_k, \cdots, f^i_K\} \in \mathbb{R}^{K \times (4+C)}$. Then the cluster are fed into a shared multilayer perceptron (MLP) and output a feature map $\hat{F}^i = \{\hat{f}^i_1, \cdots, \hat{f}^i_k, \cdots, \hat{f}^i_K\} \in \mathbb{R}^{K \times C_a}$. A maxpool layer with a Softmax function are followed to predict one-dimensional attentive weights $\{w^i_1, \cdots, w^i_k, \cdots, w^i_K\} \in \mathbb{R}^{K \times 1}$ for each neighbor point. The generated keypoint $\tilde{x}_i$ is calculated as the weighted sum of the Euclidean coordinates of the neighbor points. Denoting the Euclidean coordinates of the neighbor points as $\{x^i_1, \cdots, x^i_k, \cdots, x^i_K\} \in \mathbb{R}^{K \times 3}$, then the output keypoint $\tilde{x}^i \in \mathbb{R}^3$ of this cluster can be calculated as

$$\tilde{x}^i = \sum_{k=1}^{K} w^i_k \cdot x^i_k, \quad i = 1 \cdots M \tag{1}$$

Unlike the offset predicting method in USIP, our attentive points aggregation method ensures that the generated keypoint is within the convex hull of the input cluster. Then, each feature $\hat{f}^i_k$ is assigned with a corresponding attentive weight $w^i_k$, so we get an attentive feature map $\tilde{F}^i \in \mathbb{R}^{K \times C_a}$ as

$$\tilde{F}^i = \{\tilde{f}^i_1, \cdots, \tilde{f}^i_k, \cdots, \tilde{f}^i_K\} = \{w^i_1 \cdot \hat{f}^i_1, \cdots, w^i_k \cdot \hat{f}^i_k, \cdots, w^i_K \cdot \hat{f}^i_K\}, \quad i = 1 \cdots M \tag{2}$$

The attentive feature map is further summed to output a global feature $\tilde{f}^i \in \mathbb{R}^{C_a}$ for each cluster. After that, an additional MLP with a Softplus function are followed to predict the saliency uncertainty $\sigma^i$ for each keypoint. The output of the detector network are $M$ keypoints with corresponding saliency uncertainties and also clusters $\{F^1, \cdots, F^M\}$ with the attentive feature maps $\{\tilde{F}^1, \cdots, \tilde{F}^M\}$. Finally, we select keypoints with lower saliency uncertainties without Non-maximum Suppression (NMS).

### 3.2 Descriptor

The network structure of the descriptor is shown in the bottom row of Fig. 1. The input of the descriptor network are the random dilation clusters $\{F^1, \cdots, F^M\}$ and attentive feature maps $\{\tilde{F}^1, \cdots, \tilde{F}^M\}$ from the detector network. A cluster $F^i$ is firstly fed into a shared MLP to get individual feature for each neighbor point and then a maxpool layer is applied to obtain a $C_f$-dimensional global cluster feature. After that, the global cluster feature is duplicated and then concatenated with individual point features and the attentive feature map $\tilde{F}^i$. The concatenated feature map is further passed into another shared MLP with a maxpool layer to generate a $d$-dimensional descriptor. The attentive feature map from the detector network significantly improves the performance of the descriptor and we will illustrate the effectiveness in ablation study.

### 3.3 Loss

Denoting source and target point clouds as $\mathcal{S}$ and $\mathcal{T}$, keypoints from source and target point clouds as $\mathbf{X}^{\mathcal{S}}$ and $\mathbf{X}^{\mathcal{T}}$, the corresponding saliency uncertainties as $\mathbf{\Sigma}^{\mathcal{S}}$ and $\mathbf{\Sigma}^{\mathcal{T}}$, and descriptors as $\mathbf{Q}^{\mathcal{S}}$ and $\mathbf{Q}^{\mathcal{T}}$. The ground truth relative rotation $\mathbf{R}$ and translation $\mathbf{t}$ is also provided. For the training of detector,

we follow USIP [16] to use the probabilistic chamfer loss and point-to-point loss. Probabilistic chamfer loss is utilized to minimize the distance of keypoints in source and target point cloud and point-to-point loss is defined to penalize the keypoints for being too far from the original point clouds. Please refer to [16] for the details of probabilistic chamfer loss and point-to-point loss.

**Matching loss**　We propose matching loss to train the descriptor in a weakly supervised manner. The triplet loss in 3DFeatNet [17] samples positive and negative descriptor pairs according to the distance of two point clouds and does not utilize the geometric positions of keypoints. Unlike 3DFeatNet, the key idea of our matching loss is a soft assignment strategy which explicitly estimates the correspondences of descriptors. For each source keypoint $\tilde{x}_i^{\mathcal{S}} \in \mathbf{X}^{\mathcal{S}}$ and the corresponding descriptor $q_i^{\mathcal{S}} \in \mathbf{Q}^{\mathcal{S}}$, the Euclidean distances with descriptors of all target keypoints are calculated, which is denoted as $\mathbf{d}_i^{\mathcal{S}} = \{d_{i1}^{\mathcal{S}}, \cdots, d_{ij}^{\mathcal{S}}, \cdots, d_{iM}^{\mathcal{S}}\}$, where $d_{ij}^{\mathcal{S}} = \left\| q_i^{\mathcal{S}} - q_j^{\mathcal{T}} \right\|_2^2$. The matching score vector is denoted as $\mathbf{s}_i^{\mathcal{S}} = \{s_{i1}^{\mathcal{S}}, \cdots, s_{ij}^{\mathcal{S}}, \cdots, s_{iM}^{\mathcal{S}}\}$, which can be calculated as

$$s_{ij}^{\mathcal{S}} = \frac{\exp(\frac{1/d_{ij}^{\mathcal{S}}}{t})}{\sum_{j=1}^{M} \exp(\frac{1/d_{ij}^{\mathcal{S}}}{t})} \tag{3}$$

where $t$ is the temperature to sharpen the distribution of the matching score. Then the corresponding keypoint of $\tilde{x}_i^{S}$ can be represented as a weighted sum of all target keypoints,

$$\hat{x}_i^{\mathcal{S}} = \sum_{j=1}^{M} s_{ij}^{\mathcal{S}} \cdot \tilde{x}_j^{\mathcal{T}} \tag{4}$$

Soft assignment can be considered as an approximate derivable version of nearest neighbor search on descriptors. Intuitively, the target keypoints with more similar descriptors to the source keypoint will be given a larger score. When $t \rightarrow 0$, the soft assignment will degenerate to a deterministic nearest neighbor search. Similarly, we can also calculate the corresponding source keypoint for each target keypoint using soft assignment strategy. Furthermore, according to the saliency uncertainty $\sigma_i \in \mathbf{\Sigma}$ of each keypoint, we introduce weight for each keypoint,

$$\tilde{w}_i = M \cdot \frac{w_i}{\sum_{i=1}^{M} w_i}, \quad w_i = \max\{\sigma_{\max} - \sigma_i, 0\} \tag{5}$$

where $\sigma_{\max}$ is the pre-defined upper bound of saliency uncertainty. The final matching loss aims at minimizing the distance between estimated corresponding keypoints, which can be represented as

$$\mathcal{L}_{matching} = \sum_{i=1}^{M} \tilde{w}_i^{\mathcal{S}} \left\| \mathbf{R}\tilde{x}_i^{\mathcal{S}} + \mathbf{t} - \hat{x}_i^{\mathcal{S}} \right\|_2^2 + \sum_{i=1}^{M} \tilde{w}_i^{\mathcal{T}} \left\| \mathbf{R}\hat{x}_i^{\mathcal{T}} + \mathbf{t} - \tilde{x}_i^{\mathcal{T}} \right\|_2^2 \tag{6}$$

Intuitively, the reduction of the matching loss will motivate the soft assignment strategy to select keypoints that are closer in space as matching points, which pull the matched descriptors closer and unmatched descriptors away. Besides, the introduction of weights of keypoints makes keypoints with lower saliency uncertainties have higher weights in the matching loss.

## 4 Experiments

### 4.1 Experiment setting

**Datasets**　We evaluate our proposed RSKDD-Net on two large scale outdoor LiDAR datasets, namely KITTI Odometry Dataset [37] (KITTI dataset) and Ford Campus Vision and Lidar Dataset [38] (Ford dataset). KITTI dataset provides 11 sequences (00-10) with ground truth vehicle poses and we use Sequence 00 to train, Sequence 01 for validation and the others for testing[2]. Ford dataset contains two sequences and we only use this dataset for testing. For training, the current point cloud with the 10th point cloud after it is considered as a training pair. For testing, we use the current point cloud with the five consecutive frames before and after it as test data. Consequently, we obtain over 100,000 testing pairs in KITTI dataset and Ford dataset.

**Evaluation metric**    We follow the same evaluation metrics as in 3DFeatNet [17] and USIP [16] for keypoint detector and descriptor, namely *Repeatability*, *Precision* and *Registration performance*.

*Repeatability* is introduced in USIP to evaluate the stability of detected keypoints. Given source and target point clouds with the ground truth transformation, a keypoint in source point cloud is repeatable if its distance to the nearest keypoint in target point cloud is less than a distance threshold $\epsilon_r$. Repeatability is defined as the ratio of repeatable keypoints to all detected keypoints.

*Precision* is utilized in 3DFeatNet to jointly evaluate keypoint detector and descriptor. Given a source keypoint $\tilde{x}_i^{\mathcal{S}}$, we search the corresponding target keypoint $\tilde{x}_j^{\mathcal{T}}$ based on nearest neighbor search of descriptors. Meanwhile, the ground truth corresponding keypoint location $\bar{x}_j^{\mathcal{T}}$ in target point cloud is calculated according to the ground truth transformation. If $\tilde{x}_j^{\mathcal{T}}$ and $\bar{x}_j^{\mathcal{T}}$ is within a distance threshold $\epsilon_p$, this correspondence is considered as valid and precision is defined as the valid ratio.

*Registration performance* is evaluated using RANSAC algorithm. Followed 3DFeatNet, the number of RANSAC iterations is adjusted based on 99% confidence and capped at 10000 iterations. We evaluate the relative translation error (RTE), relative rotation error (RRE) as in 3DFeatNet. A registration is considered as successful if RTE $< 2$ m and RRE $< 5°$. Besides, we also calculate the average inlier ratio and iteration times of the RANSAC algorithm.

**Baseline algorithms**    We compare our approach with three handcrafted 3D keypoint detectors ISS [4], Harris-3D [6] and SIFT-3D [5] with handcrafted keypoint descriptor FPFH [8] and two deep learning-based 3D keypoint detectors and descriptors: 3DFeatNet [17] and USIP [16]. We use the implementation in PCL [39] for handcrafted detectors and descriptor. For USIP, we use the provided source code and retrain the model on KITTI dataset due to the lack of pretrained model of descriptors. For 3DFeatNet, we simply use the provided pretrained model and test it on KITTI dataset and Ford dataset. Besides, we also evaluate the repeatability of random sampled points for reference. Experiments on other handcrafted descriptors are displayed in our supplementary.

**Implementation details**    In the pre-processing, we firstly downsample the input point cloud by a Voxelgrid filter of 0.1 m grid size and extract the surface normals and curvature of each point as additional features following USIP [16]. Then 16384 points are randomly sampled from the downsampled point cloud as input point cloud. The dilation ratio $\alpha_d$ is set to 2 and the number of neighbor points is set to 128. The network is implemented using PyTorch [40]. We use SGD as the optimizer with learning rate of $0.001$ and momentum of $0.9$. Temperature $t$ in matching loss is set to $0.1$. We train the network in two-stage, firstly the detector is trained with probabilistic chamfer loss and point-to-point loss. Then we train the descriptor based on the pretrained detector using matching loss and the detector network will also be fine-tuned in this stage. The network is trained on NVIDIA GeForce 1080Ti and evaluated on a PC with Intel i7-9750H and NVIDIA GeForce RTX 2060.

### 4.2   Evaluation

**Efficiency**    We evaluate the efficiency of our RSKDD-Net and other two learning-based methods on KITTI dataset and the computation time is shown in Table 1. The top row of Table 1 represents the number of input points and the second row represents the number of keypoints. Thanks to the random sample strategy and no requirements for per-point saliency estimation, our method shows a much higher efficiency than the other two learning-based methods. Noting that the computation time of 3DFeatNet and USIP increases mainly with the number of input points and the number of keypoints, respectively. In comparison, the computation time of our method does not increase significantly with the number of input points and keypoints. Specifically, our method is more than $30\times$ faster than USIP and 3DFeatNet to detect 512 keypoints from 16384 input points.

**Repeatability**    We calculate the repeatability of 128, 256 and 512 keypoints with distance threshold of 0.5m. Besides, the repeatability with different distance thresholds for 512 keypoints are also evaluated for reference. The results are displayed in Fig. 3. According to the results, the repeatability of our method outperforms all other methods for 128 to 512 keypoints with a significant margin. Specifically, the repeatability of our method is about 20% higher than that of USIP for 512 keypoints at distance threshold of 0.5 m, which demonstrates the high stability of our selected keypoints.

Table 1: Computation time (ms)

| Input points | 4096 | | | 8192 | | | 16384 | | |
|---|---|---|---|---|---|---|---|---|---|
| Keypoints | 128 | 256 | 512 | 128 | 256 | 512 | 128 | 256 | 512 |
| 3DFeatNet | 66.8 | 70.8 | 81.4 | 136.2 | 156.2 | 169.9 | 367.6 | 413.7 | 420.7 |
| USIP | 76.6 | 163.6 | 296.7 | 99.6 | 171.0 | 310.9 | 115.2 | 203.4 | 378.5 |
| RSKDD-Net | **3.8** | **4.1** | **4.7** | **4.3** | **5.2** | **6.5** | **5.7** | **8.5** | **10.1** |

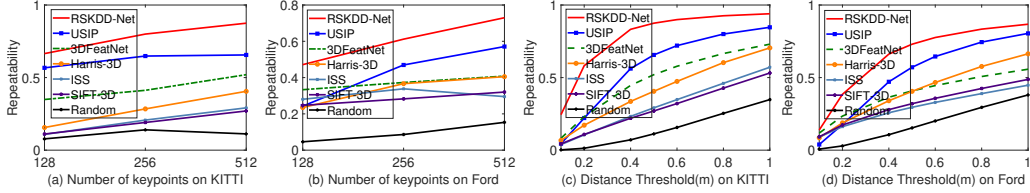

Figure 3: (a) and (b): Repeatability with different numbers of keypoints. (c) and (d): Repeatability with different distance thresholds.

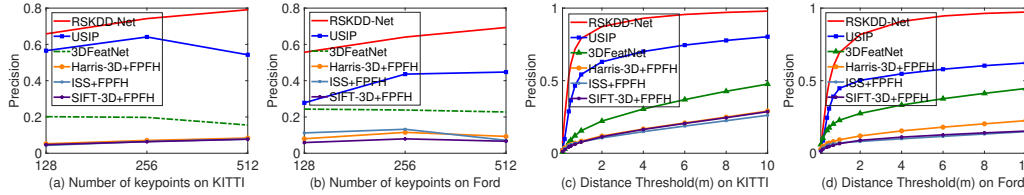

Figure 4: (a) and (b): Precision with different numbers of keypoints. (c) and (d): Precision with different distance thresholds.

**Precision**   We evaluate the precision for 512 keypoints with different distance thresholds and also the precision for different numbers of keypoints with distance threshold of 1.0 m. The results in Fig. 4 show that our RSKDD-Net provides a much higher precision than other methods for different numbers of keypoints and also different distance thresholds. Noting that our RSKDD-Net performs better on KITTI dataset than on Ford dataset, the reason may be that the point clouds in KITTI dataset are more structured so that our RSKDD-Net can detect keypoints with more geometric information.

**Registration performance**   The number of keypoints for evaluation of registration performance is fixed at 512. The experiment results are displayed in Table 2. According to the experiments, our method gives a better RTE than both handcrafted and learning-based methods. Although the RRE and success rate is slightly inferior to USIP, the inlier ratio of our method is more than twice than that of USIP and brings much less average iteration times due to the high precision and repeatability. Taken together, our method outperforms all of the other methods.

**Qualitative Visualization**   We provide several qualitative visualization results of our proposed method. Two registration results are shown in left two columns of Fig. 5. The number of keypoints is fixed to 512 and the visualization results show a large inlier ratio of our proposed keypoint detector and descriptor. Besides, we give a sample of our keypoints detection results in the right column of Fig. 5. Although the method does not explicitly remove the points on ground plane, the detected keypoints automatically avoid the ground points and concentrate on points with geometric information like facades and corners. More qualitative results are provided in our supplementary.

### 4.3   Ablation study

We provide ablation study to illustrate the effectiveness of the random dilation cluster, attentive feature map, matching loss. All experiments of ablation study are performed on KITTI dataset. The repeatability and precision are calculated with 128, 256 and 512 keypoints. The distance thresholds of repeatability and precision are fixed at 0.5 m and 1.0 m, respectively.

Table 2: Registration performance on KITTI dataset and Ford dataset

| Methods | KITTI dataset | | | | | Ford dataset | | | | |
|---|---|---|---|---|---|---|---|---|---|---|
| | RTE (m) | RRE (deg) | Inlier | Iter | Success | RTE (m) | RRE (deg) | Inlier | Iter | Success |
| Harris+FPFH | $0.38 \pm 0.33$ | $1.79 \pm 1.24$ | 0.018 | 10000 | 82.9% | $0.51 \pm 0.59$ | $0.48 \pm 0.90$ | 0.187 | 935 | 74.0% |
| ISS+FPFH | $0.59 \pm 0.39$ | $1.24 \pm 0.98$ | 0.024 | 10000 | 92.3% | $0.54 \pm 0.56$ | $0.70 \pm 1.16$ | 0.160 | 1364 | 74.4% |
| SIFT+FPFH | $0.57 \pm 0.39$ | $1.39 \pm 0.96$ | 0.040 | 9973 | 92.5% | $0.56 \pm 0.56$ | $0.68 \pm 1.11$ | 0.243 | 376 | 75.3% |
| 3DFeatNet | $0.31 \pm 0.26$ | $0.73 \pm 0.64$ | 0.093 | 6591 | 97.9% | $0.37 \pm 0.42$ | $0.61 \pm 0.73$ | 0.100 | 5642 | 91.3% |
| USIP | $0.10 \pm 0.05$ | $\mathbf{0.35 \pm 0.21}$ | 0.243 | 468 | **100**% | $0.12 \pm 0.06$ | $\mathbf{0.38 \pm 0.39}$ | 0.195 | 870 | **100**% |
| RSKDD-Net | $\mathbf{0.09 \pm 0.06}$ | $0.50 \pm 0.28$ | **0.586** | **32** | 99.9% | $\mathbf{0.11 \pm 0.08}$ | $0.58 \pm 0.41$ | **0.505** | **41** | 99.5% |

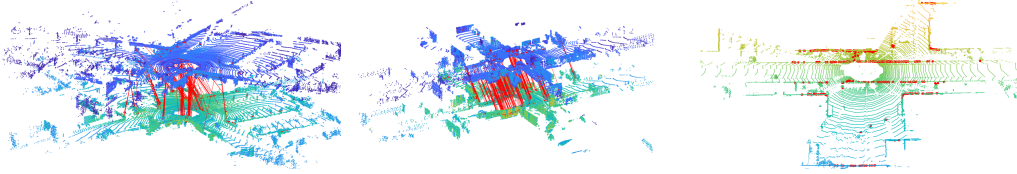

Figure 5: The left two images are two registration results and the red lines between two point clouds represent successful matchings. The right image is a sample of keypoints detection results and the red points denote the detected keypoints.

**Random dilation cluster**   Here we compare the performance of our proposed random dilation cluster with DPC [31] and the corresponding repeatability and precision are displayed in Fig. 6(a) and Fig. 6(b). The results show that the random dilation cluster significantly improves the repeatability and precision of our network. And the random dilation cluster performs similar to and in some scenes even slightly better than DPC (e.g., the precision of 128 and 256 keypoints). In addition, our method is more simple and has no requirements for neighbor points sorting, which reduces the time complexity of neighbor points searching.

**Attention feature map**   In order to demonstrate the effectiveness of the attentive feature map for learning of descriptor, we remove the attentive feature map in the descriptor network and evaluate the precision. The results can be seen in Fig. 6(c). According to the results, the introduction of the attentive feature map results in an obvious increase in precision. The precision for different numbers of keypoints increases by about 0.1 with the attentive feature map.

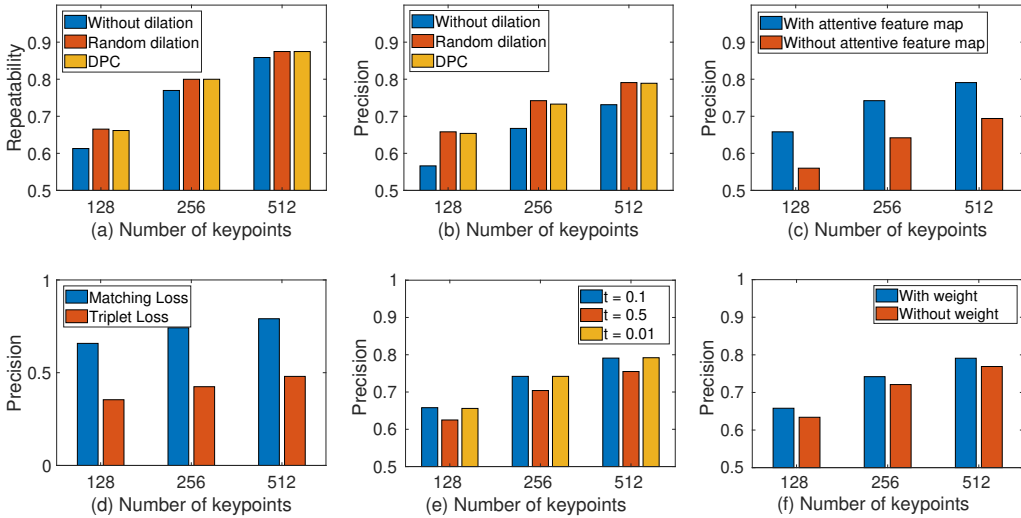

Figure 6: (a) Repeatability with different dilation strategy. (b) Precision with different dilation strategy. (c) Precision with and without attentive feature map. (d) Precision with matching loss and triplet loss. (e) Precision with difference temperature $t$. (f) Precision with and without weight.

**Matching loss** We compare our matching loss with the triplet loss used in 3DFeatNet. Following 3DFeatNet, we sample positive and negative point cloud pairs based on the distance between two point cloud and the saliency uncertainty are also included in the triplet loss. Then we use the triplet loss to replace the proposed matching loss and retrain the network. The precision of the two loss functions can be seen in Fig. 6(d). According to the experiments, our matching loss significantly outperforms triplet loss in 3DFeatNet in our settings. Specifically, the precision of our method is about twice of that of the network training with triplet loss. Besides, we also perform experiments to study the effect of the temperature $t$ and also the weight in the proposed matching loss. As shown in Fig. 6(e), the precision with $t = 0.5$ is obviously lower than the other two ones, which is due to the poor approximation of the soft assignment strategy with large $t$. And the performance will not change significantly when $t < 0.1$. According to Fig. 6(f), the introduction of weight in the matching loss improves the performance of the descriptor.

## 5 Conclusion

This paper proposes a learning-based method to jointly detect keypoints and generate descriptors in large scale point cloud. The proposed RSKDD-Net achieves state-of-the-art performance with much faster inference speed. To overcome the drawback of random sampling, we propose a novel random dilation cluster strategy to enlarge the receptive field and an attention mechanism for positions and features aggregation. We propose a soft assignment-based matching loss so that the descriptor network can be trained in a weakly supervised manner. Extensive experiments are performed and demonstrate that our RSKDD-Net outperforms existing methods by a significant margin in repeatability, precision and registration performance.

## Broader Impact

The proposed RSKDD-Net provides an efficient scheme to detect keypoints and generate descriptors for large scale point cloud registration. The method is most likely to be applied to localization and mapping system of autonomous vehicles to reduce the computation of point cloud registration, which may promote the development of autonomous driving. The development of autonomous driving can reduce the workload of human drivers and the incidence of traffic accidents, however, can have an impact on the determination of liability for traffic accidents and results in unemployment of human drivers. Besides, the proposed method has also applications on unmanned aerial vehicles. However, unmanned aerial vehicles can be utilized in military field, thereby threatening human safety. We should explore more beneficial applications of this method, such as promoting the development of autonomous driving to improve the quality of human life and improve its safety to reduce accidents.

## Acknowledgments and Disclosure of Funding

This work is funded by National Natural Science Foundation of China (No. 61906138), the European Union's Horizon 2020 Framework Programme for Research and Innovation under the Specific Grant Agreement No. 945539 (Human Brain Project SGA3), and the Shanghai AI Innovation Development Program 2018. We thank Guohao Li for helpful discussion.

## Footnotes

[2]We simply drop Sequence 08 because of the large errors of ground truth vehicle poses in this sequence

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
