[Supplementary Material · supplementary.pdf]

# Supplementary Materials

We provide more details and experiment results of the proposed RSKDD-Net in the supplementary material.

**Network details**    The channels of the MLP in RSKDD-Net are as follows:

- First MLP in detector network: [64,128,256]

- Second MLP in detector Network: [256,256,1]

- First MLP in descriptor network: [64,128,128]

- Second MLP in descriptor network: [128,128]

**Comparison with handcrafted descriptors**    We combine our keypoint detector with several handcrafted descriptors and make comparisons with our learning-based descriptors. We selected three handcrafted descriptors: FPFH, SHOT and USC. The number of keypoints is fixed at 512 and the experiments are performed on KITTI dataset. The results are displayed in Fig. 1(a). According to the results, our descriptor significantly outperforms handcrafted descriptors.

Figure 1: (a) Comparison with handcrafted descriptors. (b) Comparison with learning-based detectors.

**Comparison with learning-based detectors**    To demonstrate the effectiveness of our keypoint detectors, we combine the detector in 3DFeatNet and USIP with handcrafted descriptor and then make comparisons with ours. The number of keypoints is fixed at 512 and the experiments are performed on KITTI dataset. USC is chosen as the handcrafted descriptor due to the high performance. The results are shown in Fig. 1(b). The results show that the keypoints provided by our detector are more distinctive than that of USIP and 3DFeatNet.

**Descriptor dimension**    We analyze the effect of descriptor dimension on the performance. We evaluate the precision for descriptor dimensions of 32, 64, 128 and 256 and display the results in Fig. 2(a). According to the results, there is no significant drop even with descriptor dimension of 32 and the precision is basically unchanged after the dimension is greater than 128.

**Dilation ratio**    We perform experiments with different dilation ratio and the results is shown in Fig. 2(b). The results show that the repeatability and precision hardly increases after the dilation ratio is greater than 2.0.

**Jointly learning**    We calculated the repeatability of keypoint detector with and without fine-tuning in descriptor learning. The results can be seen in Fig. 2(c). The results show that the fine-tuning in descriptor learning also improves the performance of detector.

Figure 2: (a) The precision of different descriptor dimensions. (b) Precision and repeatability of different dilation ratios. (c) Repeatability with and without fine-tuning

**More visualization results**   We provide more visualization results of our keypoints detector and also matching results. Fig. 3 shows keypoints detection results for different number of keypoints on KITTI dataset (left) and Ford dataset (right). According to the visualization results, the keypoints selected by our method concentrate in the same region with the increase of number of keypoints.

Figure 3: Keypoints detection results with 128, 256, and 512 keypoints (top to bottom).

Figure 4: Visualization of registration results on KITTI dataset.

Figure 5: Visualization of registration results on Ford dataset.

Figure 6: Keypoints detection results on KITTI dataset.

Figure 7: Keypoints detection results on Ford dataset.