[Reviews · NeurIPS 2020]

Review 1

Summary and Contributions: This paper proposes a new method to efficiently generate keypoint detector and descriptor on 3D point clouds. They propose a few novel modules in the network such as random dilation cluster to improve the receptive field of the network, attention-based point aggregation, and a novel matching loss. The proposed method outperforms previous state-of-the-art methods in various metrics.

Strengths: The overall method is sound and reasonable. I’m very surprised that the proposed matching loss outperforms the triplet loss and to be honest still find it difficult to believe it, but the ablation study is adequate to support their claim. The proposed method may have some significance in the fields that require keypoint detection and descriptors such as 3D localization and reconstruction. The registration experiment seems to have demonstrated a promising potential. The proposed random dilation cluster may be useful for generic 3D reasoning as well.

Weaknesses: Fundamentally, the performance of the proposed method seems to be throttled by the initial random choice of cluster centers. I am worried that the proposed keypoint detection may not work well if we use too small number of keypoints compared to the size of the scene. Metric: I am surprised that recall is not a metric of keypoint detector when precision is. For keypoint detection, I would care more about missing keypoint than uninformative ones. Ablation study: There are many details of the proposed components that could better be ablated. For example, the effect of keypoint weight in the matching loss and the effect of temperature parameter on the performance may have been helpful. minor: Although the overall paper is sound as a computer vision application paper, it is a bit concerning that his paper may not fully fit the venue of NeurIPS community.

Correctness: The method and the claims is sound to me.

Clarity: The writing quality looks okay to me. The paper nicely motivates the problem and introduces the previous works to ground this paper on. I had no major problem following the details of this work. However, there were a few grammatical errors and it could better be revised. Furthermore, the paper is not fully self-contained and often refers import details to other papers.

Relation to Prior Work: This is the first paper I read in deep learning keypoint detection and descriptor. I am not aware of prior works and as far as what this paper has described, the previous state-of-the-art methods look recent and the evaluation protocols seem to match as well.

Reproducibility: Yes

Additional Feedback: N/A post rebuttal comment: Authors have addressed most of my concerns and I do not see a major weakness of this work on other reviews. I am slightly concerned about the way the runtime is evaluated in this work, though. I would like to update my rating to accept.


Review 2

Summary and Contributions: The key idea of the paper is to use random sampling to select candidate points for keypoint detection through a saliency mapping and learn the keypoint descriptors simultaneously for the task of point cloud registration. An attention mechanism is introduced to aggregate the keypoint features. The paper suggests an additional loss function for the task. The method is evaluated on 2 outdoor datasets: the KITTI dataset and the Ford dataset.

Strengths: Even though the Fig. 1 that describes the network architecture seems complicated the method is straightforward. The proposed method outperforms previous works on keypoint repeatability. The method produces comparable results on the registration task with the previous work of USIP.

Weaknesses: The method seems to predict saliency maps for all the points in the point cloud treating all points as potential keypoints. This seems computationally heavy for large point clouds. The claim on computational time does not correspond to the computational time reported on the paper of the previous work of USIP.

Correctness: In Table 1 the computational time report does not match the computational time reported in the USIP paper. It is very different. The proposed method is surprisingly fast for a method that seems to consider all keypoints as potential keypoints. (line 90)

Clarity: The paper is clearly written. The task is explained in detail and the components of the approach are described in detail.

Relation to Prior Work: The proposed approach is compared with previous works on the field and what separates the proposed method from previous works is clearly explained.

Reproducibility: Yes

Additional Feedback: ===== POST AUTHOR FEEDBACK ====== I read the other reviews and the author feedback. My concerns regarding the speed comparison and the confusing sampling definition were mostly addressed in the rebuttal. However FPS is a slow sampling algorithm I would have liked to see USIP running times with maybe random sampling or both methods compared without the sampling runtime. I am still a bit unsure about the random sampling. For me their method can compensate for the few number of keypoints as long as they sample enough points. With few sampled points everything will break. So I was hoping they would focus on that side a bit more, the authors admit the issue in the rebuttal and claim that even like this they still outperform other methods (few papers also have no issues at all). But I am still on the positive side.


Review 3

Summary and Contributions: Point could registration is an important task in self-driving and mapping resaserch area. However, previous SOTA methods adopt farthest point sample (FPS) which results in O(n^2) complexity which makes them hard to become real-time applications. This work proposes random dilation cluster strategy and attention learning mechanism with random sampling to avoid the information loss. It also has a joint detector-descriptor learning pipeline for more accurate registration. They finally achieve SOTA results in both time and performance.

Strengths: By improving the speed (15x!!!) and the performance at the same time, this paper already has enough impact to be on a top conference. The overall design makes a lot of senses, the learning of the detector-descriptor is clearly inspired by pointnet alike structure and attention mechanism.

Weaknesses: From idea, method, to experiment, they are all impressive. I can hardly find a drawback of this paper. However, it looks like the ablation against the random sampling itself is not fully analyzed. The final results only suggest that sampling was not that important as long as the receptive field is large enough. I believe the author need to discuss this more. For example, if the sampling process is purely random, then how can one guarantee the descriptor and keypoint can be matched in different maps. What if the attention mechanism is unable to correct this due to the sampled points are simply too different? I think Figure.6 is meant for this kind of problem but it's still hard to imagine such issue can be resolved that easily. I would like to know the authors' explanation.

Correctness: I think the intuition and the method are both reasonable. Abundant experiments are provided.

Clarity: The paper is clearly written in terms of high level idea, contribution, and overview. However, please detach the legend list in Fig.3,4. It's hard to read.

Relation to Prior Work: The proposed method has a very clear feature against the current SOTAs. It improves both speed and performance by consider the basic rather than complex idea. Impressive.

Reproducibility: Yes

Additional Feedback: I have read the rebuttal. I vote for acceptance. Looking forward to see the code.

[Author Response · NeurIPS 2020]

We thank all reviewers, and will incorporate all comments and suggestions in the final paper.

**[Q1] More analysis of random sampling (R1,R4):** The perfor-
mance of the method is throttled by random sampling when the
number of keypoints is small. However, the proposed modules (e.g.,
random dilation cluster, attentive points aggregation, etc.) can weaken
the negative effect of random sampling and therefore, the performance
of smaller number of keypoints can be improved by sampling more
candidate points. We performed experiments with smaller number of
keypoints and different dilation ratios on KITTI dataset to illustrate
the effect of random sampling and the receptive field on performance.
The results are displayed in Table 1. The distance thresholds for
repeatability and precision are set to 0.5 m and 1.0 m, respectively.
Note that we select keypoints based on the predicted saliency un-
certainty. Denoting the number of keypoints as $N_k$, the number of
sampled points as $N_s$ and dilation ratio as $\alpha$. In our current implemen-
tation, the number of sampled points is twice the number of selected
keypoints (e.g., $N_s = 128$ if $N_k = 64$). According to Table 1, the
performance significantly drops as $N_k$ and $N_s$ drop. Enlarging $\alpha$ can

| $N_k$ | $N_s$ | $\alpha$ | Repeatability | Precision |
|---|---|---|---|---|
| 32 | 64 | 2 | 0.301 | 0.385 |
| 32 | 64 | 4 | 0.355 | 0.451 |
| 32 | 64 | 6 | 0.360 | 0.457 |
| 64 | 128 | 2 | 0.479 | 0.528 |
| 64 | 128 | 4 | 0.537 | 0.587 |
| 64 | 128 | 6 | 0.538 | 0.587 |
| 128 | 256 | 2 | 0.665 | 0.658 |
| 128 | 256 | 4 | 0.697 | 0.688 |
| 128 | 256 | 6 | 0.696 | 0.688 |
| 32 | 128 | 2 | 0.452 | 0.509 |
| 32 | 256 | 2 | 0.627 | 0.628 |
| 32 | 512 | 2 | 0.707 | 0.691 |

Table 1: The performance of different number
of keypoints $N_k$, number of sampled points $N_s$
and dilation ratio $\alpha$.

improve the performance due to the enlargement of the coverage of the whole network. However, when $N_s$ is too small
(e.g., $N_s = 64$), simply enlarging the receptive field is hard to cover the whole point cloud and the performance is
greatly limited. Even with limitations, the proposed method achieves better performance than state-of-the-art. The given
$N_k$ (e.g, $N_k = 128$) is a reasonable number compared to the large scale point cloud. We also provide an alternative
strategy for better performance with smaller $N_k$. The high efficiency of our method permits us to sample more candidate
points for a smaller $N_k$, which does not cause a significant increasing on runtime. For example, if we need $N_k = 32$,
we can set $N_s = 256$ and only select 32 keypoints from them. As shown in the bottom three lines of Table 1, the
performance is significantly improved for small $N_k$ if we sample more candidate points. The results indicate that
sampling method is not a primary constraint on performance when the coverage of the network is large enough.

**[Q2] Metrics of keypoint detection (R1):** We agree that recall should be considered as an evaluation metric to evaluate
the keypoint detector. However, due to the lack of ground truth for keypoint detector, it is intractable to define the recall.
Nonetheless, precision measures the performance of keypoint detector comprehensively, it relates to the repeatability,
informativeness of generated keypoints and the effectiveness of the descriptor. We think the metrics provided in the
paper are sufficient to evaluate the performance of the proposed keypoint detector and descriptor.

**[Q3] More ablations (R1):** We performed ablation studies on the
weight and temperature $t$ in the proposed matching loss. As shown
in Table 2, the introduction of weight in matching loss improves the
performance of the descriptor. Precision with different $t$ is shown
in Table 3. The soft assignment can not represent nearest neighbor
search well if $t$ is too large (e.g., $t = 0.5$). In our implementation,
$t = 0.1$ is a proper choice and the performance will not change

| Number of keypoints | 128 | 256 | 512 |
|---|---|---|---|
| With weight | 0.658 | 0.742 | 0.791 |
| Without weight | 0.634 | 0.721 | 0.769 |

Table 2: Precision with and without weight in
matching loss.

significantly when $t < 0.1$. Based on the suggestions of the reviewer, we will add more ablations in the final paper.

**[Q4] Reported runtime of USIP (R2):** The runtime reported in paper of USIP does not include the time of farthest
point sampling (FPS) and the calculation of descriptor. The time-consuming FPS is implemented in the dataloader
according to the released code of USIP and they only reported the processing time of the detector network itself. Thus,
we re-calculate the runtime including FPS and descriptor generation using the released code on our own platform.

**[Q5] Saliency estimation (R2):** We do not estimate saliency for all points in
the point cloud. Instead, we only randomly sample several candidate points and
use the proposed random dilation cluster as well as an attention mechanism to
aggregate neighbor points and estimate the saliency. Thus, saliency estimation
is only performed on sampled candidate points rather than all points. There
exists minor writing typos of the denotations in line 90: we generate keypoints
$\mathbf{X} \in \mathbb{R}^{M \times 3}$ and saliency uncertainties $\mathbf{\Sigma} \in \mathbb{R}^M$ rather than $\mathbf{X} \in \mathbb{R}^{N \times 3}$ and
$\mathbf{\Sigma} \in \mathbb{R}^N$, where $M$ is much smaller than $N$. We will correct it in the final paper.

| $t$ | 128 | 256 | 512 |
|---|---|---|---|
| $t = 0.1$ | 0.658 | 0.742 | 0.791 |
| $t = 0.5$ | 0.625 | 0.704 | 0.755 |
| $t = 0.01$ | 0.656 | 0.742 | 0.792 |

Table 3: Precision with different
temperature $t$ in matching loss.

**[Q6] Concerns of too different sampled points (R4):** The performance of the proposed method is stable when the
coverage of the network is large enough (see **[Q1]** in this rebuttal document). The attentive mechanism tends to generate
informative points in its receptive field and the network gives high weights to stable and informative keypoints. With a
large coverage, the detection results cover most of the informative points, which are stable and consistent in different
point clouds. Thus, the network can generate consistent keypoints even the sampled points in two point clouds are very
different. As shown in the bottom three lines of Table 1, even with only 32 keypoints, our network achieves a high
repeatability with a large $N_s$, which indicates the stability of the detected keypoints in different point clouds.

[Meta-Review · NeurIPS 2020]

The initial scores for this paper were: 6: Marginally above the acceptance threshold. 6: Marginally above the acceptance threshold. 8: Top 50% of accepted NeurIPS papers. A very good submission; a clear accept. While the reviewers are overall positive, they raised several concerns regarding: Technical details of the method. Metrics used for evaluation. More thorough ablations. Computational time. The authors provide a rebuttal, which addresses most of the weak points. In the post-rebuttal discussion, and considering the rebuttal, R1 upgrades their score from 6 to 7. The final scores are 6, 7, 8. The AC is convinced by the positive arguments of the reviewers and recommends Accept. The authors are strongly encouraged to take into account all reviewers' feedback and final recommendations when preparing the final version.